# In Vitro Viral Recovery Yields under Different Re-Suspension Buffers in Iron Flocculation to Concentrate Viral Hemorrhagic Septicemia Virus Genotype IVa in Seawater

**DOI:** 10.3390/ani13050943

**Published:** 2023-03-06

**Authors:** Na-Gyeong Ryu, Eun-Jin Baek, Min-Jae Kim, Kwang-Il Kim

**Affiliations:** Department of Aquatic Life Medicine, Pukyong National University, Busan 48513, Republic of Korea

**Keywords:** iron flocculation, oxalic acid, ascorbic acid, viral hemorrhagic septicemia virus, viral genome recovery, viral infective recovery

## Abstract

**Simple Summary:**

Environmental DNA (eDNA) has attracted attention as a monitoring tool in the aquaculture industry to detect aquatic viral diseases in seawater. To be able to use the low concentration of viruses in seawater, an additional concentration process needs to be employed. In this study, the iron flocculation method was applied to concentrate the viral hemorrhagic septicemia virus (VHSV), and the recovery rate was estimated to determine the suitability of two re-suspension buffers, oxalic and ascorbic acid. It was determined that, while both of these buffers could be efficient for recovering viral genome copy, the oxalic acid buffer was more suitable for preserving viral infectivity than the ascorbic acid buffer. Thus, the iron flocculation method using an oxalic acid buffer could be efficient for evaluating actual viral transmission and is expected to predict the occurrence of diseases in the environment.

**Abstract:**

Iron flocculation is widely used to concentrate viruses in water, followed by Fe-virus flocculate formation, collection, and elution. In the elution stage, an oxalic or ascorbic acid re-suspension buffer dissolved iron hydroxide. After the concentration of viral hemorrhagic septicemia virus (VHSV) in seawater (1 × 10^1^ to 1 × 10^5^ viral genome copies or plaque-forming unit (PFU)/mL), the recovery yield of the viral genome using quantitative real-time PCR (qRT-PCR) and viral infectivity using the plaque assay were investigated to evaluate the validity of the two re-suspension buffers to concentrate VHSV. The mean viral genome recovery yield with oxalic and ascorbic acid was 71.2 ± 12.3% and 81.4 ± 9.5%, respectively. The mean viral infective recovery yields based on the PFU were significantly different between the two buffers at 23.8 ± 22.7% (oxalic acid) and 4.4 ± 2.7% (ascorbic acid). Notably, although oxalic acid maintains viral infectivity over 60% at a viral concentration above 10^5^ PFU/mL, the infective VHSVs were not sufficiently recovered at a low viral concentration (10^2^ PFU/mL, <10%). To support this result, concentrated VHSV was inoculated in Epithelioma papulosum cyprini (EPC) cells to confirm cell viability, viral gene expression, and extracellular viral titer. All results demonstrated that oxalic acid buffer was superior to ascorbic acid buffer in preserving viral infectivity.

## 1. Introduction

Environmental DNA (eDNA), which is genetic material extracted from environmental samples, including water, can be applied as a pathogen monitoring tool [1,2]. Despite the applicability of eDNA for disease surveillance in aquaculture, the low concentration of viral particles released from diseased aquatic animals in seawater makes it difficult to directly use water samples, in their natural state, in general, using molecular methods (e.g., polymerase chain reaction) without a concentration process. Among the several concentration methods, such as polyethylene glycol precipitation [3], ultracentrifugation [4], tangential flow filtration [5], and the negatively charged membrane method [6], viral particle coagulation has been applied to recover viral particles from water [7,8,9,10]. In the coagulation method, various filters and metal substances, such as aluminum, sodium alginate, and sodium silicate, are used as coagulants to increase the efficiency of viral recovery yields by combining viruses with membrane filters [8,11]. Notably, the iron flocculation method, based on ferric chloride as a coagulant, is widely used because of the high recovery yields of viral particles [12]. Iron flocculation consists of three steps: Fe-virus colloid formation, collection, and elution. Since viral particles can be released under different water conditions (freshwater or seawater), several factors, including Fe concentration, pH, salinity, stirring speed, alkalinity, collection filter type, and re-suspension buffer, should be considered [7,12,13,14,15,16,17]. Notably, the elution step involving the dissolution of iron hydroxide by a re-suspension buffer is not essential to detect viruses by molecular assays [18,19]; the re-suspension buffer employed in the step is closely associated with the recovery of virus infectivity [14]. Thus, based on the purpose of this study, it is necessary to evaluate the viral recovery yield under experimental conditions.

Viral hemorrhagic septicemia virus (VHSV), an enveloped negative-sense single-stranded RNA virus (family *Rhabdoviridae*, genus *Novirhabdovirus*), is the most serious pathogen causing mass mortality in both freshwater and marine fish, resulting in significant economic loss [20,21]. VHSV can be horizontally transmitted through water containing the virus released from the infected fish [22,23]. In Korea, since the first outbreak of viral hemorrhagic septicemia (VHS) in olive flounder (*Paralichthys olivaceus*) in Pohang in 2001 [24], it has been reported annually as an endemic disease [25]. Notably, in aquaculture systems, infective viral particles in seawater emitted from diseased fish could be a source of reinfection and disease outbreaks. In a previous study [26], even though viral genomic recovery yields of VHSV (genotype IVa) were 23.0% using a negatively charged membrane, recovery yields of viral infectivity were not determined. Thus, since the analysis of infective viral particles is a crucial factor in determining the distributional transmission and interaction with susceptible species to aquatic viral disease, both quantitative genomic analysis and analysis of infective viral particles in seawater are needed for surveillance. Therefore, to apply iron flocculation as a diagnostic procedure for disease surveillance, viral genomic and infectious recovery yields should be determined.

In this study, the viral recovery yields of VHSV were compared between two re-suspension buffers (oxalic acid and ascorbic acid) in the iron flocculation process. Viral genomic recovery was determined based on a quantitative real-time PCR (qRT-PCR) assay. Viral infective recovery was analyzed using a plaque assay on EPC cells.

## 2. Materials and Methods

### 2.1. VHSV Culture and Titration

#### 2.1.1. Virus Culture

Epithelioma papulosum cyprini (ECACC, Cat No. 93120820; EPC) cells were cultured in Leibovitz’s medium (L-15; Gibco, Grand Island, NY, USA) supplemented with 10% (*v*/*v*) fetal bovine serum (FBS; Gibco, Grand Island, NY, USA) and 1× antibiotic–antimycotic solution (GenDEPOT, Katy, TX, USA) at 25 °C. VHSV 21OfPh isolate (genotype IVa) [27] from diseased olive flounder was propagated in an L-15 medium containing 5% FBS at 15 °C for 5 days.

#### 2.1.2. Plaque Assay

Following the appearance of the cytopathic effect (CPE), the viral titer of the supernatant was determined using a plaque assay. Briefly, EPC cells (approximately 4 × 10^6^ cells/well) seeded in 6-well plates were inoculated with 200 μL of 10-fold serially diluted supernatants for 2 h at 15 °C. After the washing of the cells with phosphate-buffered saline (PBS; Gibco, Grand Island, NY, USA), a semi-solid overlay medium consisting of 2% (*w*/*v*) SeaPlaque ^TM^ GTG ^TM^ agarose (Lonza, Rockland, WA, USA) was added. Following the appearance of visible plaques (4–5 days post inoculation) which could be visually and sufficiently counted as individual plaques, cells were fixed and stained with 25% formaldehyde and 0.5% crystal violet for 4 h. Then, the semi-solid medium was carefully removed, followed by washing the residue with tap water, and the number of plaques was counted.

#### 2.1.3. Quantitative Real-Time PCR

Quantification of the viral cDNA copies using primers and TaqMan probes targeting a fragment of the VHSV nucleoprotein (*N*) gene was performed by qRT-PCR [28]. Briefly, qRT-PCR was performed with a reagent mixture containing 10 μL of HS Prime Q-Master Mix (2X, Real-time PCR for TaqMan Probe; GenetBio, Daejeon, Korea), 800 nM of each primer (forward and reverse; Table 1), 200 nM of the probe, 0.4 μL of 50X ROX dye (GenetBio, Daejeon, Korea), 6 μL of nuclease-free water, and 0.8 μL of cDNA using the StepOne real-time PCR system (Applied Biosystems, Waltham, MA, USA). The standard curve was generated using recombinant plasmid DNA harboring the VHSV nucleoprotein (*N*) gene (82 bp) cloned based on the pGEM T-vector (Promega GmbH, Mannheim, Germany). To determine the cut-off value for quantitative analysis, 95% of the limit of detection (LOD_95%_) was set as described by Uhlig et al. [29]. Following serial two-fold dilutions of plasmid DNA (from 40 copies μL^−1^ to 0.3125 copies μL^−1^), qRT-PCR was conducted with 12 replicates. LOD_95%_, which indicates 95% of a positive result at the lowest number of viral DNA copies and confidence intervals, was decided through probit regression analysis using MedCalc^®^ (version 20.015).

### 2.2. Concentration of VHSV in Seawater by Iron Flocculation

#### 2.2.1. Iron Flocculation Process

To compare VHSV particle recovery between viral genomic and infective particles by iron flocculation, VHSV particles in seawater were concentrated with minor modifications as described previously [12]. First, before the pre-filtration process, seawater was stored at room temperature (approximately 20 °C) for 1 h to precipitate debris, and pre-filtered using a vacuum pump (Gast, Benton Harbor, MI, USA) with a glass microfiber filter (GF/A; pore size, 1.6 μm; Whatman, Maidstone, UK). If water flux was declined by membrane fouling, the membrane filter was replaced with a new filter. Then, 50 μL of iron chloride solution [4.83 g of iron (III) chloride hexahydrate (FeCl_3_·6H_2_O) per 100 mL of distilled water] was added to 500 mL of serially diluted VHSV-spiked seawater (1 × 10^1^ to 1 × 10^5^ viral genome copies/mL, 1 × 10^1^ to 1 × 10^5^ PFU/mL) and gently mixed using a magnetic stir bar and stir plate (<120 rpm) for 1 h at 20 °C to allow Fe-VHSV flocculate formation. The Fe-VHSV flocculate was collected on a polycarbonate (PC) membrane filter (pore size, 1.0 μm; Whatman, Maidstone, UK) holder with a polyethylene sulfone (PES) membrane filter (pore size, 0.8μm; Whatman, Maidstone, UK) using a peristaltic pump (<12 psi; Eyela, Japan). Following collection of the Fe-VHSV flocculates, the membrane was transferred to a 5 mL round-bottom tube, and 1 mL of re-suspension buffer (oxalic or ascorbic acid buffer; pH 6.0 ± 0.2, as previously described by Kim et al. [14]) was added. Viral re-suspension was performed for 2 h using a Bio RS-24 Mini-Rotator (30 rpm; 170 Biosan, Riga, Latvia) in a dark room at 4 °C.

#### 2.2.2. Comparison of Viral Genome Copies and Infective Viral Particle Recovery

To compare the viral genomic recovery yields using different re-suspension buffers in the iron flocculation process, total RNA from the concentrate of VHSV-spiked seawater (200 µL) was extracted using the yesR™ Total RNA Extraction Mini Kit (GenesGen, Busan, Korea). The extracted RNA exhibited an absorbance ratio between 2.0 and 2.2 when measured at A_260/280 nm_ with a NanoVue Plus Spectrophotometer (GE Healthcare, Chicago, IL, USA) used for the cDNA template. RNA was reverse-transcribed using the PrimeScript™ 1st cDNA Synthesis Kit (Takara, Kusatsu, Japan) according to the manufacturer’s instructions. The number of viral cDNA copies in the concentrate was determined using qRT-PCR, as described in Section 2.1.3. Moreover, the viral infective recovery yields under different re-suspension buffers were compared using the plaque assay described in Section 2.1.2. The recovery yield of viral genome copies was calculated based on the total viral cDNA copies (genome copies/mL) of VHSV-spiked seawater and concentration products (recovered viral cDNA copies per mL × volume of concentration product/spiked viral cDNA copies per mL × volume of seawater). The recovery yield of infective particles was calculated based on the total spiked infective particles (PFU/mL) in seawater and concentration products (recovered PFU per mL × volume of concentration product/spiked PFU per mL × volume of seawater). The experiments were performed in triplicate, and the results of the recovery rate (%) were presented as the mean ± standard deviation (SD). The positive results of qRT-PCR were determined using a cut-off value of LOD_95%_. Statistical analyses were conducted using a two-way analysis of variance (ANOVA) with GraphPad Prism software (version 9.5.0), and statistical significance was set at *p* < 0.05.

### 2.3. Assessment of Infectivity Recovery for Viral Concentrates Eluted with Different Buffers

In Experiment 1 (Exp. 1), to clarify the infectivity recovery yield under different re-suspension buffers, the viability and morphology of EPC cells inoculated with the concentrated VHSV-spiked seawater were compared. The EPC cells (approximately 5 × 10^5^ cells/well) seeded into 24-well plates were inoculated with 100-fold diluted concentrates (originally, VHSV-spiked seawater at 1 × 10^1^ to 1 × 10^5^ PFU/mL) at 15 °C. After 2 h of viral adsorption, followed by washing with PBS, VHSV-concentrate-inoculated cells were incubated with fresh L-15 medium containing 5% FBS for 48 h. Cell viability was determined using the Cellrix^®^ Viability Assay Kit (Medifab, Seoul, Korea) according to the manufacturer’s instructions. Briefly, EPC cells were reacted with the reagent at 15 °C for 3 h, and absorbance was measured at 450 nm using an Infinite^®^ 200 microplate reader (Tecan, Männedorf, Switzerland). Cell viability compared to naïve EPC cells was calculated using a two-way ANOVA with GraphPad Prism software (ver. 9.5.0). Statistical significance was set at *p* < 0.05.

In Experiment 2 (Exp. 2), to compare the viral gene expression of the concentrate eluted by different re-suspension buffers, the expression levels of *N* and glycoprotein (*G*) genes were investigated in concentrate-inoculated EPC cells. After inoculation with the concentrate under the same experimental conditions as in Exp. 1, EPC cells were incubated for 48 h with fresh L-15 medium containing 5% FBS. The relative expression levels of the VHSV *N* and *G* genes were determined by qRT-PCR. Briefly, total RNA was extracted from concentrate-inoculated cells using the yesR™ Total RNA Extraction Mini Kit (GenesGen, Busan, Korea), followed by cDNA synthesis using the PrimeScript™ 1st cDNA Synthesis Kit (Takara, Kusatsu, Japan). For gene expression analysis, qRT-PCR was carried out using the StepOne real-time PCR system (Applied Biosystems, Waltham, MA, USA) with the Prime Q-Master Mix (2X, real-time PCR with SYBR Green I; GenetBio, Daejeon, Korea). The reaction was performed in a final volume of 20 μL, containing 10 μL of 2X Prime Q-Master Mix, 1 μL of each primer, 7 μL of distilled water, and 1 μL of cDNA. The primers and qRT-PCR conditions used are presented in Table 1. The relative quantitation values of the amplified genes in triplicate samples were measured using the 2^−ΔΔCt^ method as described by Rao et al. [32]. Statistical analysis was conducted using a two-way ANOVA with GraphPad Prism software (version 9.5.0), and significance was supported by a *p*-value < 0.05.

In Experiment 3 (Exp. 3), the viral cDNA copies of the extracellular virus from concentrate-inoculated EPC were compared under the same experimental conditions in Exp.1. Following incubation of concentrate-inoculated EPC cells for 48 h, the supernatant was collected, and viral cDNA copies in each triplicate well were determined by qRT-PCR, as described in Section 2.1.3. Statistical analyses were conducted using a two-way ANOVA with GraphPad Prism software version 9.5.0. Statistical significance was set at *p* < 0.05.

## 3. Results

### 3.1. Determination of the LOD_95%_ Value for qRT-PCR

To support the reliability of qRT-PCR by differentiating false-positive from non-specific reactions, the LOD_95%_ value was determined in 12 replicates with low viral copies of pDNA harboring *N* gene fragments. From the standard curve (Figure 1A), the correlation coefficient (*R*^2^) of the plotted points (viral pDNA copies from 3.17 × 10^1^ to 3.17 × 10^8^ copies/μL) was 0.9975. The LOD_95%_ value was 9.57 pDNA copies/μL (95% confidence interval, 4.807–40.106 pDNA copies/μL; Figure 1B). Therefore, the cut-off value for qRT-PCR was set at 9.57 copies/μL.

### 3.2. Genomic and Infective Recovery Yields Using Iron Flocculation from VHSV-Spiked Seawater

Viral recovery yields were found to be higher as more VHSVs were present in the seawater. Based on the viral cDNA copies (spiked VHSV copies at 1 × 10^5^–1 × 10^1^ viral cDNA copies/mL of seawater), the mean ± SD viral genomic recovery yields by ascorbic acid (81.4 ± 9.5%) were superior to oxalic acid (71.1 ± 12.3%) (Table 2). Meanwhile, based on the PFU (spiked VHSV copies at 1 × 10^5^–1 × 10^1^ PFU/mL of seawater), the mean (± SD) viral infective recovery yields by oxalic acid and ascorbic acid were significantly different at 23.8 ± 22.7% and 4.4 ± 2.7%, respectively (Table 3). Notably, even though VHSVs were spiked at a concentration of 1 × 10^1^ to 1 × 10^2^ PFU/mL in seawater, infectious particles were not identified in concentrates eluted with ascorbic acid.

### 3.3. Assessment of Viral Infectivity Recovery at Different Re-Suspension Buffers

Based on a comparison of cell viability and morphology (Exp. 1), the relative viability of oxalic-acid-resuspended concentrates inoculated on the EPC cells was significantly decreased compared to that of ascorbic-acid-resuspended concentrates (Figure 2A). Moreover, the typical CPE caused by VHSV infection, including retracted and rounded refractile cells, were observed to be more pronounced in oxalic acid groups at viral concentrations ranging from 1 × 10^3^ to 1 × 10^5^ PFU/mL (Figure 2B). Regarding viral gene expression (Exp. 2), the expression levels of both the VHSV *G* and *N* genes were significantly upregulated in the oxalic acid group than in the ascorbic acid group in a concentration-dependent manner (Figure 3A,B). Furthermore, viral cDNA copies of the extracellular virus (Exp. 3) showed a significant difference (approximately a 12 to 531-fold difference) between the oxalic and ascorbic acid groups at spiked concentrations of 1 × 10^2^ to 1 × 10^5^ PFU/mL (Figure 3C).

## 4. Discussion

With a low concentration of virus particles in the water below the detection limit, a general molecular assay can be difficult to apply without a concentration process. Among the virus concentration methods, iron flocculation is broadly applied to concentrate aquatic animal viruses in water, such as red sea bream iridovirus (RSIV) [33], tilapia tilapinevirus (TiLV) [19], cyprinid herpesvirus-2 (CyHV-2) [17], and white spot syndrome virus (WSSV) [14]. Notably, several conditions in the iron flocculation process are associated with the viral recovery yield and preservation of infectivity. Herein, to investigate the applicability of surveillance of VHSV in seawater, the viral genome and infective recovery yields were assessed in vitro using different re-suspension buffers (oxalic and ascorbic acid) in the iron flocculation process.

There are a variety of VHSV genotypes (I, II, III, and IV). To identify their genotypes and genetic variants, nucleotide sequence analysis is needed. qRT-PCR developed by Garver et al. [28] has been recommended as a validated method sufficient for VHSV surveillance in all life stages of susceptible species [34]. Based on the LOD_95%_ and the reliability of positive results of at least 95%, a cut-off value of qRT-PCR described by Garver et al. [28], 9.57 pDNA copies/μL were served (Figure 1B). Accordingly, quantitative results below the cut-off were not served as valid data in this study. In terms of viral genome recovery yield, viral cDNA copies eluted by oxalic and ascorbic acid were recovered over 70% without significant differences (Table 2). In our previous study [14], the viral genomic recovery yields of WSSV as a DNA virus also showed no significantly different results between re-suspension buffers. Thus, re-suspension buffer containing oxalic and ascorbic acid for eluting concentrates in iron flocculation might not be associated with genomic recovery yields.

Viral infectivity is an important factor as one of the criteria for assessing actual viral transmission in disease surveillance or prediction. Of note, between oxalic- and ascorbic-acid-eluted concentrate, the recovery of viral infectivity in oxalic acid remarkably appeared above VHSV-spiked concentration greater than 1 × 10^5^ PFU/mL (>60%; Table 2) and was different from findings in the ascorbic acid group (<10%). Notably, at a low viral concentration (<1 × 10^2^ PFU/mL), the infective VHSVs were slightly recovered by oxalic acid (<10%). In contrast, infective VHSVs were not recovered in ascorbic acid.

As the proteins or lipids in serum can protect the surface of viral particles from damage, several previous studies have used serum sources (e.g., FBS) during viral concentration or storage to preserve viral infectivity [35,36]. However, in the iron flocculation experiment in this study, the infective recovery yields were not improved by the addition of FBS (exhibited less than 3.9% of recovery yield at 0.1 and 1% FBS addition in either seawater or re-suspension buffer; data not shown). Since the recovery yield of the iron flocculation method is influenced by the formation of the Fe-virus flocculate and its dissolution during the re-suspension process [14,33], the macromolecular substances present in FBS might interfere with the formation of the flocculate (spiked in seawater) and re-suspension process (spiked in oxalic acid buffer).

The EPC cell viability in the oxalic acid group was significantly lower than that in the ascorbic acid group (Figure 2). Moreover, the levels of viral gene expression and extracellular viral cDNA copies in oxalic acid were significantly higher than those in ascorbic acid, supporting the infectivity preservation of concentrates eluted by oxalic acid in the iron flocculation process (Figure 3). pH is one of the most important parameters that influence the preservation of viral infectivity, as strong acidic (approximately <pH 3) or basic (>pH 12) conditions could damage viral particles. Meanwhile, a previous study exhibited that VHSV could maintain infectivity between pH 6 and 9 for 24 h [37]. Based on these findings, the pH levels (approximately 6.0) of the re-suspension buffers used in this study were confirmed to be sufficient for viral re-suspension without affecting VHSV infectivity. In iron flocculation, ascorbic acid advances the dissolution of iron hydroxide by reducing Fe^3+^ to Fe^2+^ to allow chelation with EDTA [12]. In more detail, when reducing the Fe^3+^ to Fe^2+^, ascorbic acid enhances the production of radicals by the Fe^2+^/EDTA/H_2_O_2_ reaction complex. These radicals are generated by the one-electron oxidation of the respective reduced compounds, which promotes an additional oxyradical-mediated chain reaction [38,39]. Even though ascorbic acid, known as vitamin C, likely seems to be associated with improved EPC cell viability, oxygen-free radicals produced by it might reduce the ability to preserve infectivity during the elution step in iron flocculation, consistent with a previous study [14]. Oxalic acid advances the dissolution of iron hydroxide by binding directly to iron hydroxide to release trivalent Fe^3+^, which prevents re-precipitation and allows chelation with EDTA during iron flocculation [12,40]. Moreover, oxalic acid is used as a reductant to remove surface-adsorbed ^55^Fe from phytoplankton cells and other particles, reacting with EDTA by chelation [41]. Thus, as EDTA could inactivate the virus, oxalic acid reacting with EDTA might prevent virus inactivation during re-suspension.

## 5. Conclusions

In conclusion, this study evaluated the applicability of iron flocculation to the concentration of VHSV-IVa particles in seawater by comparing two re-suspension buffers (oxalic and ascorbic acid buffers). The viral genomic recovery yields were not significantly different between re-suspension buffers. However, although oxalic acid conserves helped to recover VHSV-IVa infectivity better than ascorbic acid at viral concentrations above 10^4^ PFU/mL (oxalic acid, >35%; ascorbic acid, <10%), the infective VHSVs were not sufficiently recovered at a low viral concentration (<1 × 10^2^ PFU/mL; oxalic acid, <10%; ascorbic acid, no recovery). This result was supported by an in vitro assessment of EPC cells inoculated with concentrates in terms of PFU recovery yield, cell viability, viral gene (*G* and *N*) expression, and extracellular virus titer. Nevertheless, for disease surveillance and prediction in aquaculture, it is essential to evaluate the validity of Fe flocculation in natural environmental samples.

## Figures and Tables

**Figure 1 animals-13-00943-f001:**
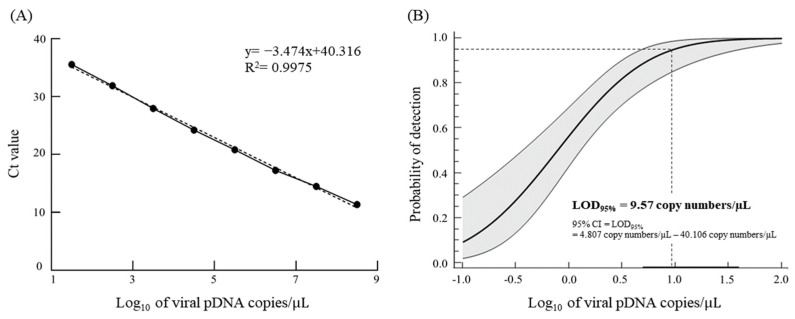
Quantification of viral cDNA was performed by VHSV specific qRT-PCR using primers and a TaqMan probe, and the cut-off value was set based on LOD_95%_. (**A**) Standard curve indicating a linear relationship between Ct (cycle number of threshold) values and the serial ten-fold pDNA (3.17 × 10^1^ to 3.17 × 10^8^ copies/μL) containing the *N* gene fragment. (**B**) The probability of detection (POD) curve was indicated with a solid line, and a 95% confidence interval was highlighted as the gray zone. LOD_95%_ (9.57 viral pDNA copies/μL) was denoted using a dotted line.

**Figure 2 animals-13-00943-f002:**
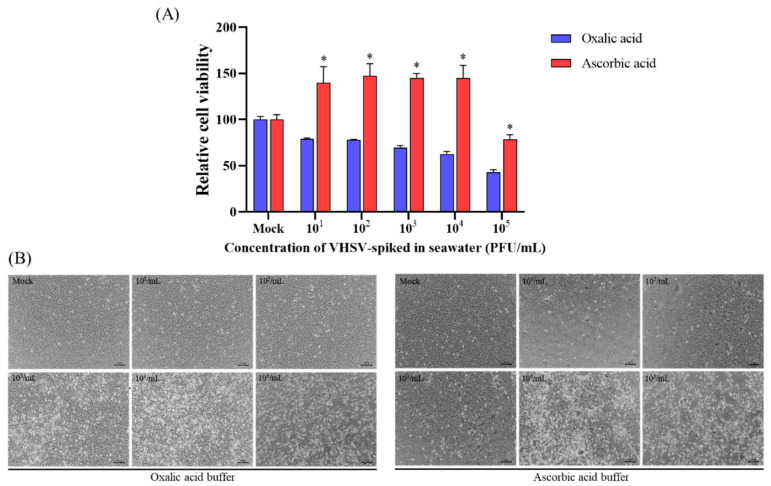
Comparison of cell viability and morphology in EPC cells inoculated by VHSV concentrates. VHSV-spiked seawater (1 × 10^1^ to 1 × 10^5^ PFU/mL) was concentrated using the oxalic or ascorbic acid buffer, and after 48 h post-infection in EPC cells, the cell viability and morphology were analyzed. Statistical analysis was performed via a two-way ANOVA (*, *p* < 0.05) using GraphPad Prism software version 9.5.0. (**A**) Relative cell viability (means ± standard deviation, SD). (**B**) The morphology of EPC cells 48 h post-infection.

**Figure 3 animals-13-00943-f003:**
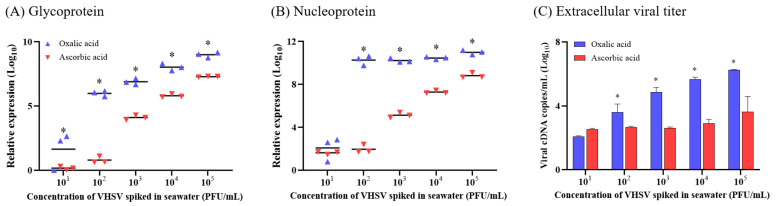
Comparison of the preserved infectivity of concentrates resuspended in oxalic and ascorbic acid buffers. Concentrates from VHSV spiked seawater (1 × 10^1^ to 1 × 10^5^ PFU/mL) were inoculated in EPC cells, and 48 h post-infection, viral gene expression levels in EPC cells and total viral cDNA copy number in the supernatant were quantified using qRT-PCR (means ± SD). A significant difference was found via two-way ANOVA (*, *p* < 0.05) using GraphPad Prism version 9.5.0. (**A**) *G* gene expression. (**B**) *N* gene expression. (**C**) Extracellular viral titer (viral cDNA copies/mL).

**Table 1 animals-13-00943-t001:** Primers uses in this study.

Purpose	Primer		Primer Sequence (5′-3′)	Condition	Product Size (bp)	Reference
Quantification	VHSV (*N* gene)	TaqMan real-time	F-ATGAGGCAGGTGTCGGAGG R-TGTAGTAGGACTCTCCCAGCATCC	50 °C, 2 min; 95 °C, 10 min (95 °C, 15 s; 60 °C, 1 min) × 40	82	Garver et al. [28]
		Probe	5′-FAM-TACGCCATCATGATGAGT-BHQ1-3′			
Gene expression	EPC cell	β-actin	F-GCTATGTGGCTCTTGACTTCGA R-CCGTCAGGCAGCTCATAGCT	94 °C, 10 min (94 °C, 20 s; 58 °C, 1 min) × 35	85	Yang et al. [30]
	VHSV	Glycoprotein	F-AACTGTCTCCAAAGAAGTGTGT R-GCCATCAAGGAGATAATGTG	95 °C, 30 s (95 °C, 15 s; 60 °C, 15 s; 72 °C, 15 s) × 40	94	Zhang et al. [31]
		Nucleoprotein	F-TTGGAGAACTGCAACACTTCAC R-CGGTCAGGATGAAGGCGTAG		82	

**Table 2 animals-13-00943-t002:** VHSV genomic recovery yield from artificial virus-spiked seawater by qRT-PCR.

Re-Suspension Buffer	Spiked Viral cDNA Copy (Viral cDNA Copies/mL)	Recovered Viral cDNA Copy (Viral cDNA Copies/mL)	Genomic Recovery Yield (%) ^a^
Oxalic acid	1.0 × 10^5^	4.3 × 10^7^ ± 7.4 × 10^6^	87.7 ± 14.8
	1.0 × 10^4^	3.4 × 10^6^ ± 1.5 × 10^5^	68.3 ± 30.0
	1.0 × 10^3^	3.5 × 10^5^ ± 5.8 × 10^4^	70.6 ± 11.4
	1.0 × 10^2^	2.9 × 10^4^ ± 1.6 × 10^4^	58.0 ± 32.0
	1.0 × 10^1^	N.D. ^b^	-
	Mean of recovery yield		71.2 ± 12.3
Ascorbic acid	1.0 × 10^5^	4.7 × 10^7^ ± 4.2 × 10^6^	95.2 ± 8.4
	1.0 × 10^4^	3.8 × 10^6^ ± 4.7 × 10^5^	76.4 ± 9.4
	1.0 × 10^3^	4.0 × 10^5^ ± 8.1 × 10^4^	79.5 ± 16.3
	1.0 × 10^2^	3.7 × 10^4^ ± 3.8 × 10^3^	74.4 ± 7.6
	1.0 × 10^1^	N.D. ^b^	-
	Mean of recovery yield		81.4 ± 9.5

^a^ Recovery (%) = (Recovered viral cDNA copies per mL × vol/spiked viral cDNA copies per mL × vol) × 100. ^b^ ND, not detected or undetermined based on the LOD_95%_ for qRT-PCR.

**Table 3 animals-13-00943-t003:** VHSV infective particle recovery yield from artificial virus-spiked seawater by plaque assay.

Re-Suspension Buffer	Spiked PFU (PFU/mL)	Recovered PFU (PFU/mL)	PFU Recovery Yield (%) ^a^
Oxalic acid	1.0 × 10^5^	3.2 × 10^7^ ± 2.0 × 10^6^	64.0 ± 4.0
	1.0 × 10^4^	1.8 × 10^6^ ± 1.3 × 10^5^	35.5 ± 2.5
	1.0 × 10^3^	4.1 × 10^4^ ± 6.1 × 10^3^	8.1 ± 1.2
	1.0 × 10^2^	2.7 × 10^3^ ± 2.9 × 10^2^	5.5 ± 0.6
	1.0 × 10^1^	3.0 × 10^2^ ± 1.1 × 10^2^	6.0 ± 2.2
	Mean of recovery yield		23.8 ± 22.7
Ascorbic acid	1.0 × 10^5^	3.6 × 10^6^ ± 3.9 × 10^5^	7.2 ± 0.8
	1.0 × 10^4^	2.6 × 10^5^ ± 1.6 × 10^4^	5.2 ± 0.3
	1.0 × 10^3^	4.3 × 10^3^ ± 1.0 × 10^3^	0.9 ± 0.2
	1.0 × 10^2^	N.D. ^b^	-
	1.0 × 10^1^	N.D. ^b^	-
	Mean of recovery yield		4.4 ± 2.7

^a^ Recovery (%) = (Recovered PFU per mL × vol/spiked PFU per mL × vol) × 100. ^b^ ND, undetermined based on the plaque assay.

## Data Availability

The data presented in this study are available upon request from the corresponding author.

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
