# Peer review of "In Vitro Viral Recovery Yields under Different Re-Suspension Buffers in Iron Flocculation to Concentrate Viral Hemorrhagic Septicemia Virus Genotype IVa in Seawater"

_animals, 2023, doi:10.3390/ani13050943_

Round 1
Reviewer 1 Report
I was invited to review this manuscript entitled "In vitro Viral Recovery Yields under Different Re-suspension 2 Buffers in Iron Flocculation to Concentrate Viral Hemorrhagic 3 Septicemia Virus Genotype IVa in Seawater" by Ryu et al. Authors described an the association of resuspension buffer for viral enrichment after iron flocculation method. Author compared the elution buffers, oxalic and ascorbic acids, used for elution of the VHSV, in terms of viral genomic recovery and validity of viral infectivity. Overall, the manuscript has well-written and it sounds interest for particular research area. I have only minor suggestion for overall description including
1. Introduction: All is fine for me but it would be nicer to address the question why this topic is particular interest
2. Discussion and overall texts: Authors only investigate the recovery of the VHSV properties and do not do the other virus or varying conditions ie, varying pH, salinity, etc. I do respect with the conditions; however, authors discussed your finding and it represents for all virus (even if it was addressed in the topic) such as the conclusion "In conclusion, this study demonstrated that the two re-suspension buffers (oxalic and ascorbic acid buffers) used in the iron flocculation method showed no significant difference in viral genomic yield,....". It seems that your investigation only delt with the VHSV but in conclusion and overall are over-statemented as it presents in not particular the VHSV. I do recommend authors to specify only the findings and may give the caution as it only investigate in an only one virus with the same external conditions.
Author Response
In accordance with your comments, we made our best efforts to improve the quality of the manuscript. We fully checked and revised the manuscript not only to emphasize the particular interest of our research but also to revise our conclusion, specifying that it is based on this research to avoid overstatement of our results.
Please find the attached file.

Reviewer 2 Report
The authors aim to investigate a monitoring tool in the aquaculture industry to detect aquatic viral diseases in seawater. They checked the effect of two re-suspension buffers, oxalic and ascorbic acid in the iron flocculation method. They demonstrate that oxalic acid buffer was superior to ascorbic acid buffer in preserving viral infectivity. The study is interesting but there are some minor question that must be addressed:
Major concerns:
In the paragraph:“2.2. Concentration of VHSV in seawater by iron flocculation”
Line 120“First, seawater pre-filtration was performed using a vacuum 120 pump (Gast, Michigan, USA) with a glass microfiber filter (GF/A; pore size, 1.6 μm; Whatman, Maidstone, UK).”
You sure seawater can directly go through the microfiber filter without pretreatment?
Standard protocol for seawater collection should be detailed.
Author Response
Thanks for your comment. We agree with you completely. We dedicated our best efforts to improving the MS to clarify the methods that adequately described our work. Next, we revised the MS following your comment as the attached file.

Reviewer 3 Report
Direct detection of VHSV in saltwater and in freshwater environments by molecular methods and for viable virus will be useful around the globe. The authors describe the iron flocculation method and comparative resuspension buffers for real-time PCR and plaque assay.
I really enjoyed reading this well-written manuscript with only minor comments made on details. I have worked on traditional concentration methods for a similar Novirhabdovirus and think the authors have shown the utility here for VHSV. Certainly, they have shown the efficiency is high for qPCR, yet the viability of these enveloped RNA viruses can be problematic at low concentrations without the addition of a protecting agent like FBS or a cheaper alternative horse serum. I suggest these tests would benefit from adding 0.1 to 1 % serum to the water/virus and see if the viable virus would stay up at the 60% range instead of dropping for diluted VHSV. As it is, you have shown that oxalic acid is better when at high concentrations of virus (or lower serum levels that were present in the media stock). But is similarly about 5-7% recovery as was ascorbic acid, though only detected at higher virus levels. I am hopeful that this small modification of adding serum keeps the yield from dropping as in Table 3. The Batts and Winton 1989 publication (https://doi.org/10.1139/f89-125) might be included in references since they used serum addition to stabilize this fish rhabdovirus and had good eluant recovery of infectious virus.
The abstract uses the best-case scenario high virus levels for comparison and not the mean of the 10-fold dilutions that were reported in the Results section.
About line 89 should mention the inoculum volume added to wells. Also, clarify why you fixed and stained the cells when they were "faintly visible" instead of waiting a couple more days when they might be "highly visible" days 6 or 7?
I noticed in Table 1 that you have the wrong Gene Expression reverse primer for the Nucleoprotein (copied the forward primer twice) instead of the reverse primer as in the Quantification purpose.
Line 162 needs to use superscripts for the PFU/mL.
Line 218 should clarify if standard deviation or standard error?
Based on Table 2, I would think ascorbic acid is superior though not "significantly" better.
As stated earlier, I believe that addition of a serum source could have protected the virus from losing viability as seen in Table 3. Potentially redo this part of experiment, but that might depend on what other reviewers have to say. A decrease in efficiency was evident in the 10-fold lowering of virus levels. Hopefully you can keep it up above 50% recovery to aid in detection of the lowest virus possible in field studies. They might be useful to include here, though I understand that they may have been excluded for a separate publication.
Line 271 - wording choice for "served".
Viable virus is valuable to detect, as is sequencing to determine if similar or unique strain of VHSV is detected. Cannot do that with universal primers in qPCR (Garver), though a very sensitive method.
I didn't see anything about pH for the solutions, if they are important or not? Could add to methods section. Low pH is bad for fish rhabdoviruses for extended periods. Nice to know if they were the same or could have caused the difference for viability efficiency.
Author Response
Thanks for your comments. We agree with you completely. We dedicated our best efforts to improving the MS to clarify the methods and results that adequately described our work. Next, we revised the MS following the point-by-point responses as described below. Unfortunately, we tried an additional experiment using a serum source (i.e. FBS) to protect infectious viral particles but confirmed that FBS was not effective in preserving infective viral particles with the iron flocculation method. As you are probably aware, the factors related to concentrating the viral particles in water are numerous and diverse, and substantial time is needed to establish the protocols for FBS addition. Despite additional experiments being performed, the results that you expected were not achieved. Please take into consideration our difficulties due to the long time required for establishing and proving the improved protocol. As per your other valuable comments, we have thoroughly revised the manuscript. Please find the attached file.
